# Efficacy and safety of creatine phosphate sodium in the treatment of viral myocarditis: A systematic review and meta-analysis

Li Wang[1], Su-Fang Chen[2], Xi-Wei Huang[3], Zhi-Yang Wu[1], Qing-Mei Zhang[4]*

1 Department of Cardiology, Quanzhou Traditional Chinese Medicine Hospital, Quanzhou, Fujian Province, China, 2 Department of Cardiology, The 910th Hospital of the Joint Logistics Support Force of the Chinese People's Liberation Army, Quanzhou, Fujian Province, China, 3 Department of Emergency Medicine, Puning People's Hospital, Jieyang City, Guangdong Province, China, 4 Department of Pediatrics, Quanzhou First Hospital, Quanzhou, Fujian Province, China

☯ These authors contributed equally to this work.

* xbxbzqm@163.com

## Abstract

### Purpose

To systematically evaluate the efficacy and safety of creatine phosphate sodium in the treatment of viral myocarditis, and to provide guidance for its clinical treatment.

### Methods

We conducted a search of The Cochrane Library, PubMed, EMbase, and Web of Science databases to retrieve randomized controlled trials (RCTs) on the use of creatine phosphate sodium (CPS) in the treatment of viral myocarditis. The search was conducted up to April 2024. After screening the literature, extracting data, and evaluating the risk of bias in the included studies, we performed a meta-analysis using RevMan 5.4 and Stata17.0 statistical software.

### Results

A total of 104 articles were retrieved, and 9 articles with a combined total of 1,116 patients were ultimately included in the meta-analysis. The results of the meta-analysis indicated that the overall efficacy rate in the phosphocreatine sodium treatment group was higher than that in the control group [RR = 1.22, 95%CI (1.15, 1.28), P<0.00001]. Furthermore, post-treatment levels of cardiac troponin I [MD = 0.1, 95%CI (0.07, 0.13), P<0.00001] and creatine kinase isoenzyme [MD = 9.43, 95%CI (7.04,11 .82), P<0 .00001] in the phospho-creatine treatment group were lower compared to those in the control group; both differences between groups were statistically significant. Additionally, there was no significant difference observed in adverse reaction incidence between the phosphocreatine sodium treatment group and conventional treatment group [RR = 1 .07, 95% CI (0 .68, 1 .67), P = O .77].

**Data Availability Statement:** All relevant data are within the paper and its Supporting Information files.

**Funding:** This study was funded by the 2024 Municipal Traditional Chinese Medicine Key Clinical Specialty Construction Project (No. 2024 (269)). The funders had no role in study design, data collection and analysis, decision to publish, or preparation of the manuscript.

**Competing interests:** The authors have declared that no competing interests exist.

## Conclusion

Creatine phosphate sodium treatment can significantly improve the therapeutic effect of patients with viral myocarditis, and can reduce the levels of cTnI and CK-MB. Compared with conventional treatment, it has good safety.

## 1. Introduction

Viral myocarditis (VMC) is an inflammatory disease caused by viral infections of the myocardial tissue, most commonly associated with pathogens such as Coxsackievirus, parvovirus B19, and SARS-CoV-2 [1–3]. These viruses not only inflict direct damage to the myocardium but also trigger secondary immune responses. VMC can affect individuals across all age groups, with a predilection for children and young to middle-aged adults [4]. While most cases are self-limiting and resolve without specific intervention, a subset of patients experience persistent myocardial inflammation, leading to chronic dilated cardiomyopathy. In severe cases, VMC can result in acute complications such as heart failure, cardiogenic shock, and arrhythmias, which significantly increase the risk of sudden cardiac death [5–9].

Current clinical management of VMC primarily emphasizes symptomatic and supportive therapies. Among specific treatments, intravenous immunoglobulin (IVIG) is widely used for its immunomodulatory effects, particularly in patients with heightened immune activation. Studies suggest that IVIG may mitigate inflammation by modulating immune responses, potentially benefiting select patient populations; however, its efficacy remains inconsistent across different cohorts [10–12]. Interferons, known for their antiviral activity, have also been explored, especially in cases linked to certain viral infections. However, the limited efficacy and significant side effects, such as flu-like symptoms and bone marrow suppression, have restricted their adoption as a standard therapy [7,10–13].

Creatine phosphate sodium (CPS) is a high-energy phosphate compound that stabilizes myocardial cell membranes and reduces oxidative stress by supplying high-energy phosphate bonds [14–19]. Unlike the current non-specific treatments for VMC [20], CPS directly targets myocardial cells to protect their function and minimize injury. This mechanism differentiates CPS from immune-modulating agents like IVIG and antiviral treatments such as interferons, offering a distinct and potentially complementary therapeutic pathway. Clinical studies have shown that CPS may improve myocardial biomarkers and enhance cardiac function in VMC patients [21–24]. Despite these promising findings, comprehensive, large-scale, and multicenter trials evaluating CPS in VMC are still lacking.

This study aims to systematically review and analyze existing clinical evidence on the use of CPS in the treatment of VMC. By evaluating its efficacy and safety, we seek to provide robust evidence-based support for CPS as a promising adjunctive therapy for VMC.

## 2.Methods

This systematic review and meta-analysis followed the Preferred Reporting Items for Systematic Review and Meta-Analysis (PRISMA) Statement [25]. The study was prospectively registered in the international database of prospectively registered systematic reviews (PROSPERO CRD42024582920).

## 2.1 Search strategy

We updated our electronic search in PubMed, Web of Science, Embase, and the Cochrane Library through April 30, 2024. All studies were retrieved from these databases without restrictions during the online search. The Medical Subject Headings (MeSH) terms and keyword search terms used were ("Myocarditis" OR "Myocarditides" OR "Carditis") AND ("Phosphocreatine" OR "Creatine Phosphate" OR "Phosphorylcreatine" OR "Creatine Phosphate" OR "Phosphate, Creatine" OR "Neoton" OR "Phosphocreatine, Disodium Salt" OR "Disodium Salt Phosphocreatine").

## 2.2 Inclusion criteria

The criteria for inclusion of studies were as follows:

(1) Randomized controlled trials (RCTs) comparing the clinical efficacy of conventional treatment (CT) alone versus CT combined with CPS in VMC;

(2) Patients diagnosed with VMC based on clinical manifestations, myocardial injury markers, etiological detection, and imaging examinations such as electrocardiogram and echocardiography;

(3) Intervention measures: 1. The control group received CT alone, while the observation group received CT combined with CPS and/or other drugs. 2. The control group was given a low dose of CPS, and the observation group was given a high dose of CPS;

(4) The primary outcome measure of treatment effective rate had to be reported at the same time, including at least one secondary outcome indicators cardiac troponin I (cTnI), creatine kinase isoenzyme (CK-MB), adverse reaction rate.

## 2.3 Exclusion criteria

The exclusion criteria for studies were as follows:

(1) Replications of published studies;

(2) Studies with vague or missing data, data that could not be converted or combined, or key data that were unobtainable after communication with the authors;

(3) Studies for which the full text was not accessible.

## 2.4 Study selection

First, the studies were imported into EndNote software, and duplicate records were automatically removed. The remaining studies were then comprehensively screened by two researchers (Chen, S.F. and Huang, X.W). Initially, they screened the titles and abstracts of the studies. Subsequently, they evaluated the full text of eligible abstracts to select studies for inclusion in the analysis. The screening process was performed independently by the two researchers, and discrepancies were resolved by consensus among all the authors.

Finally, two researchers (Wu, Z.Y. and Zhang, Q.M.) independently extracted the required data, and a final review was conducted by an additional researcher (LW). Any uncertainties were resolved through mutual agreement among the authors. The extracted information included the first author, time of publication, age, sex ratio of participants, sample size, interventions, course of treatment, outcome indicators, and other data used to assess the risk of bias and quality of evidence. Whenever possible, missing data were obtained by contacting the study authors via email.

## 2.5 Quality assessment

The risk of bias assessment was independently conducted by two reviewers (Chen, S.F. and Wu, Z.Y.) using the Cochrane Handbook for Systematic Reviews of Interventions' tool for

assessing the risk of bias in randomized controlled trials (RCTs). Disagreements were resolved through consensus-based discussions. The degree of bias risk (low, unclear, high) was evaluated across seven domains: random sequence generation, allocation concealment, blinding of researchers and participants, completeness of outcome data, blinded assessment of outcomes, selective reporting of results, and other potential sources of bias. These evaluations were used to reflect the overall quality of each included study.

## 2.6 Outcome measures

The primary outcome was the effective rate of treatment, while the secondary outcomes included cTnI, CK-MB, and the incidence of adverse reactions after treatment. The efficacy criteria were as follows: significant effect (complete resolution of clinical symptoms), remission (improvement in clinical symptoms), and no effect (no improvement in clinical symptoms). The total efficacy rate was calculated as follows: (number of cases with significant effect + number of cases with remission) / total number of cases. The total efficacy rate was used to represent the overall curative effect.

## 2.7 Statistical analysis

RevMan software (version 5.4 for Windows; Cochrane Collaboration, Copenhagen, Denmark) and Stata statistical software (version 17.0 for Windows; Computer Resource Center, Rochester, United States) were used to analyze the relevant data from the included studies. Categorical data were analyzed using relative risk (RR) as the effect index, while continuous data were analyzed using mean difference (MD) as the effect index. The results were presented as point estimates with 95% confidence intervals (CIs).

Statistical heterogeneity among studies was assessed using the chi-square test and $I^2$ statistic. When $P \geq 0.05$ and $I^2 \leq 50\%$, no heterogeneity was indicated, and a fixed-effect model was used to pool the effect sizes. When $P < 0.05$ or $I2 > 50\%$, heterogeneity was suggested, and a random-effects model was applied. Significant clinical heterogeneity was addressed through subgroup analysis or sensitivity analysis. If neither was feasible, descriptive analysis was performed. Meta-analysis results were displayed using forest plots.

A funnel plot (RevMan 5.4) was generated to evaluate publication bias. Egger's test (STA-TAMP-64) was used to assess funnel plot asymmetry, with $P > 0.05$ indicating no publication bias.

# 3.Results

## 3.1 Literature search results

A total of 104 relevant articles were identified through searches across four databases. After removing 34 duplicate articles, 70 articles remained. By screening titles and abstracts, 55 articles, including reviews, systematic reviews, animal studies, and non-article publications, were excluded, leaving 15 articles. Finally, after full-text review, six articles that did not meet the inclusion criteria were excluded. Thus, nine studies were included in the analysis. A total of 9 studies were included. The literature screening process and results are presented in S1 Fig. A total of 1116 VMC patients were included, and their basic characteristics are summarized in Table 1. The results of the risk of bias assessment are shown in S2 Fig.

## 3.2 Outcomes

**3.2.1 Clinical efficacy of creatine phosphate sodium for viral myocarditis.** A total of nine studies [21–24,26–30] compared the treatment effective rate between the two groups in

**Table 1. The characteristics of the 9 included studies.**

| Study | Year | Age (years) (Ob/Co) | No. participants (Ob/Co) | Gender (male/female) (Ob/Co) | Interventions | Course of treatment | Outcome measure |
|---|---|---|---|---|---|---|---|
| Meng Wang (1) | 2024 | 8.05 ± 1.97 / 8.23 ± 2.09 | 40 / 40 | 17/23 / 21/19 | Medium-dose sodium creatine phosphate combined with immunoglobulin | 2 weeks | ①;②;③;④; |
| Meng Wang (2) | 2024 | 8.48 ± 1.96 / 8.23 ± 2.09 | 40 / 40 | 18/22 / 21/19 | High-dose sodium creatine phosphate combined with immunoglobulin | 2 weeks | ①;②;③;④; |
| Shaoli Lin | 2023 | 7.09 ± 3.06 / 7.0 ± 1.9 | 23 / 23 | 16/7 / 14/9 | Sodium creatine phosphate combined with sodium fructose diphosphate | 2 weeks | ①;④; |
| Jinghui Li | 2021 | 7.58 ± 2.16 / 7.64 ± 2.23 | 62 / 59 | 35/27 / 34/25 | Sodium creatine phosphate combined with immunoglobulin | 2 weeks | ①;②;③;④; |
| Qingsheng Feng | 2022 | 7.45 ± 0.65 / 7.19 ± 0.53 | 95 / 95 | 49/46 / 49/46 | Sodium creatine phosphate | 2 weeks | ①; |
| Xiao Yu | 2022 | 19.00 ± 10.00 / 19.95 ± 9.00 | 80 / 75 | 42/38 / 40/35 | Sodium creatine phosphate combined with captopril | 1 month | ①;②;③;④; |
| Ying Huang | 2019 | 7.62 ± 1.53 / 7.44 ± 1.32 | 80 / 80 | 46/34 / 44/36 | Sodium creatine phosphate combined with immunoglobulin | 2 weeks | ①;②;③;④; |
| Xiaohui Cong | 2019 | 6.13 ± 2.34 / 6.21 ± 2.6 | 43 / 43 | 24/19 / 28/18 | Sodium creatine phosphate combined with ribavirin | 2 weeks | ①;②;③;④; |
| Yunjia Li | 2018 | 6.95 ± 1.32 / 6.91 ± 1.27 | 48 / 48 | 28/20 / 25/23 | Sodium creatine phosphate combined with ribavirin | 2 weeks | ①;②;③; |
| Ziqian Wang | 2013 | 4.5 ± 3.6 / 4.9 ± 3.8 | 70 / 72 | 38/32 / 37/35 | Sodium creatine phosphate | 2 weeks | ②;③; |

Notes: Ob = Observation group; Co = Control group; Outcome measures: ①; = Effective rate of treatment; ②; = Cardiac troponin I (cTnI); ③; = Creatine kinase MB (CK-MB); ④; = Adverse reactions.

patients with VMC. Because $I^2 = 0$, P = 0.52, there was no significant heterogeneity, so the fixed effects model was used to combine the effect sizes. The results of Meta-analysis showed that the total effective rate in the observation group was significantly higher than that in the control group [RR = 1.22, 95%CI (1.15,1.28), P<0.00001, Fig 1A]. A funnel plot was used to analyze publication bias in the included studies. As shown in Fig 1B, the distribution of points on both sides of the funnel is generally concentrated in the lower part and appears asymmetrical. Egger's test for funnel plot asymmetry yielded a P-value of 0.041, indicating the presence of potential publication bias in the included studies.

**3.2.2 Meta-analysis of myocardial injury markers after treatment cTnI, CK-MB.** cTnI levels were compared between the two groups in six studies [21,24,27–30], while CK-MB levels were analyzed in seven studies [21,24,26–30]. Significant heterogeneity was observed among the studies (cTnI, $I^2$ = 91%, P < 0.0001; CK-MB, $I^2$ = 93%, P <0.0001), and therefore, a random-effects model was used for analysis. As cTnI and CK-MB are key biomarkers for myocardial injury in VMC patients [31], they were used to assess the therapeutic efficacy of CPS. Cardiac troponin I (cTnI) is a highly specific biomarker for myocardial injury, with elevated levels typically reflecting myocardial cell damage and inflammation. CK-MB, a myocardial-specific isoenzyme of creatine kinase, is released during acute myocardial injury, particularly in cases caused by viral infections. In this study, the observation group demonstrated significantly lower levels of cTnI and CK-MB compared to the control group, indicating reduced myocardial injury and improved treatment outcomes [32–36]. The results of the meta-analysis showed that the levels of cTnI and CK-MB in the observation group were significantly lower than those in the control group after treatment. [cTnI, MD = 0.1, 95%CI (0.07,0.13), P < 0.00001; CK-MB, MD = 9.43,95%CI (7.04,11.82), P < 0.00001, Fig 2A and 2B]. Sensitivity analysis of cTnI and CK-MB levels between the two groups were performed after treatment. The data were log-transformed using a random-effects model, and individual studies were

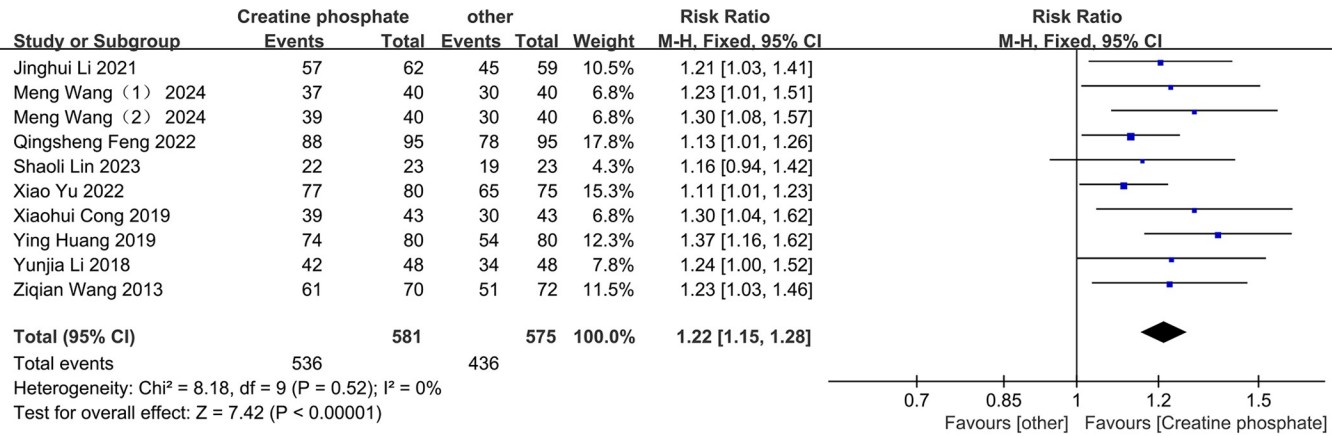

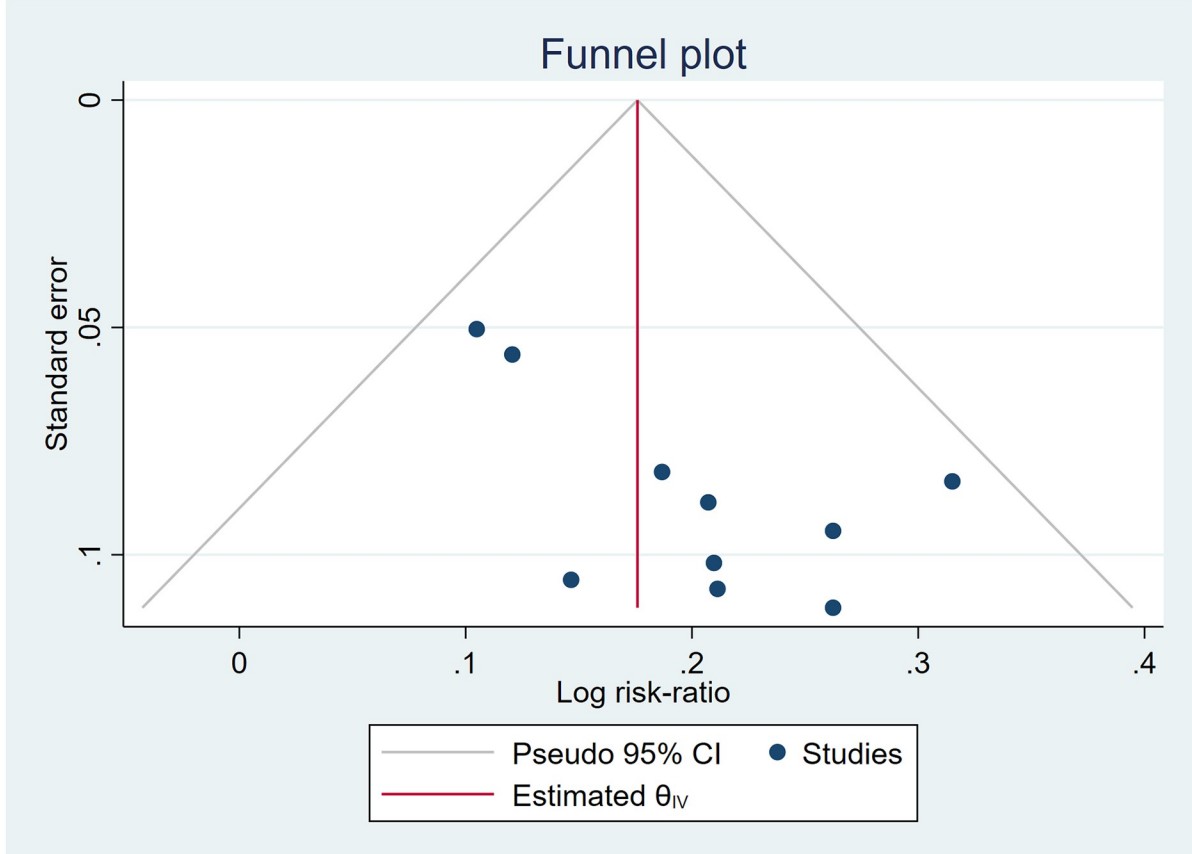

**Fig 1. Meta-analysis of the efficacy of creatine phosphate sodium in the treatment of viral myocarditis.** Forest plot(A), and funnel plot(B).

excluded one by one. The combined results indicated that the heterogeneity in cTnI could be attributed to the studies by Wang Meng and Xiao Yu, while the heterogeneity in CK-MB could be attributed to the studies by Ying Huang and Ziqian Wang. Funnel plots for cTnI and CK-MB in the randomized controlled trials were analyzed for publication bias. As shown in Fig 2C and 2D, the scatter distribution on both sides of the funnel plot was asymmetrical. Egger's test further confirmed the asymmetry, with P-values of 0.0011 for the cTnI funnel plot and 0.0005 for the CK-MB funnel plot. These results suggest potential publication bias in the included studies, which may be related to the small sample sizes or low quality of some included studies.

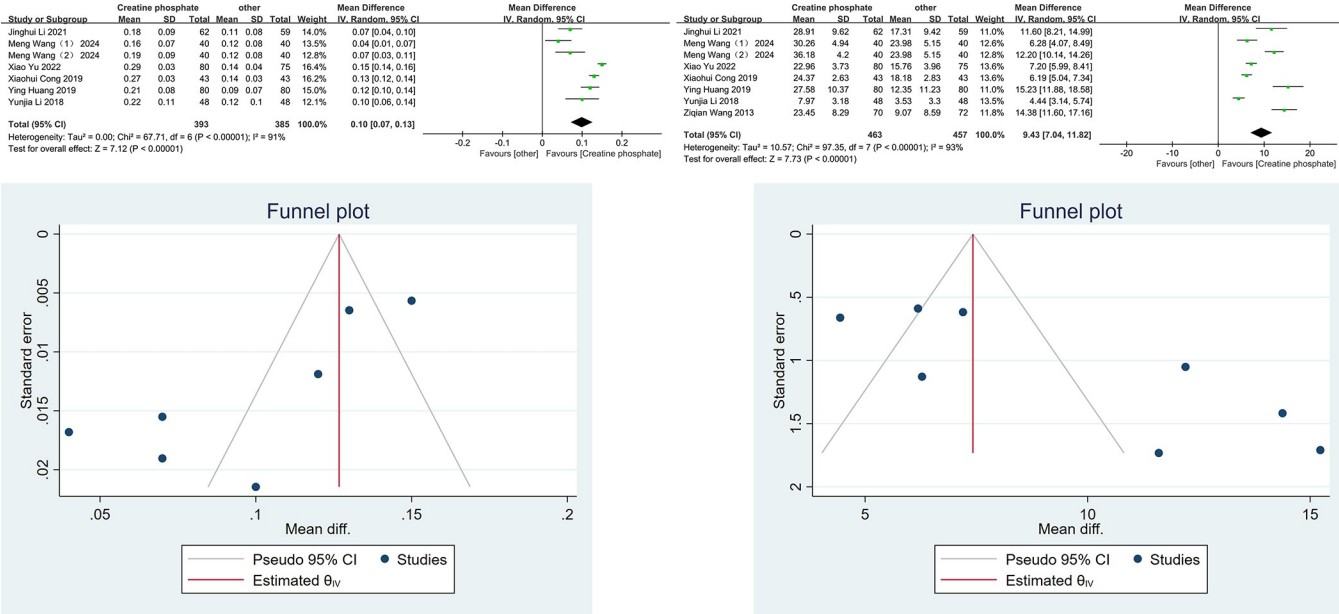

**Fig 2. Meta-analysis of myocardial injury marker levels after creatine phosphate sodium treatment.** Forest plot of cTnI (A), Forest plot of CK-MB(B), funnel plot of cTnI(C) and funnel plot of CK-MB(D).

**3.2.3 Sodium phosphocreatine adverse reactions for the treatment of viral myocarditis.** There were 5 articles comparing the adverse reactions between the two groups [21,23,24,28,30]. Because $I^2 = 0$, P = 0.80, there was no significant heterogeneity, so the fixed effects model was used to combine the effect sizes. The results of the meta-analysis showed no significant difference in the incidence of adverse reactions between the observation group and the control group [RR = 1.07, 95%CI (0.68,1.67), P = 0.77, Fig 3A]. A funnel plot was used to analyze the publication bias of this study. As shown in Fig 3B, the scatter points on both sides of the funnel plot were approximately symmetrical. Egger's test for funnel plot symmetry yielded a P-value of 0.2849, indicating that there was no publication bias in this study.

# 4.Discussion

The term myocarditis was first coined by Jean-Nicolas Corvisart in the early 19th century. Myocarditis is currently considered to be caused by acute and chronic inflammation of cardiac muscle cells, leading to associated diseases of myocardial edema and myocardial injury or necrosis [37]. Viral infection is considered to be the most common cause of myocarditis [38,39]. Although the pathogenesis of viral myocarditis is not fully understood, it has been confirmed that viral myocarditis is directly related to the virus, host genetic background, immune response, oxidation and other factors [20,40–46]. Among them, immune-mediated inflammation and lipid peroxidation induced by reactive oxygen species play an important role in myocardial cell injury [47–49].

Numerous studies have demonstrated the cardioprotective effects of CPS [50–53]. This research undertook a systematic review and meta-analysis to assess the efficacy and safety of CPS in managing VMC. Using systematic database searches, nine studies were included in the meta-analysis. Key indicators such as overall response rate, cTnI, and CK-MB levels were evaluated to determine the therapeutic efficacy of CPS, while adverse event incidence assessed its safety. Findings highlighted that the CPS treatment group showed significant improvements in myocardial biomarkers and therapeutic outcomes compared to the control group, without a

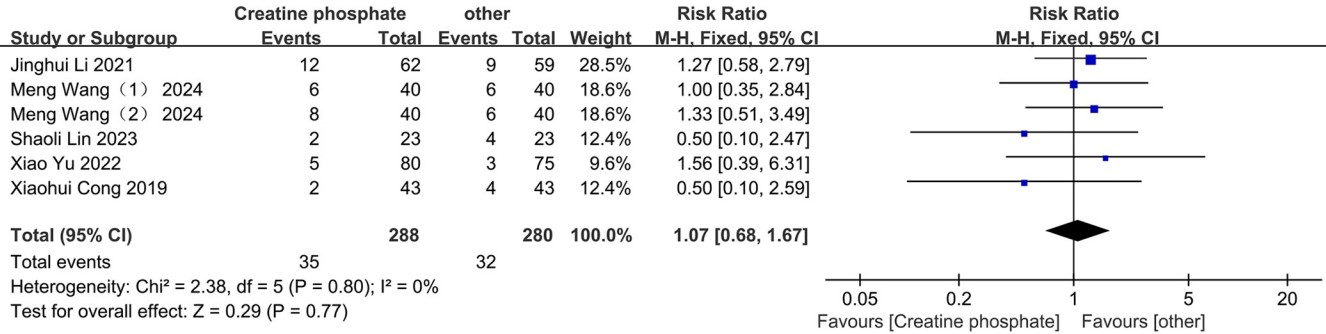

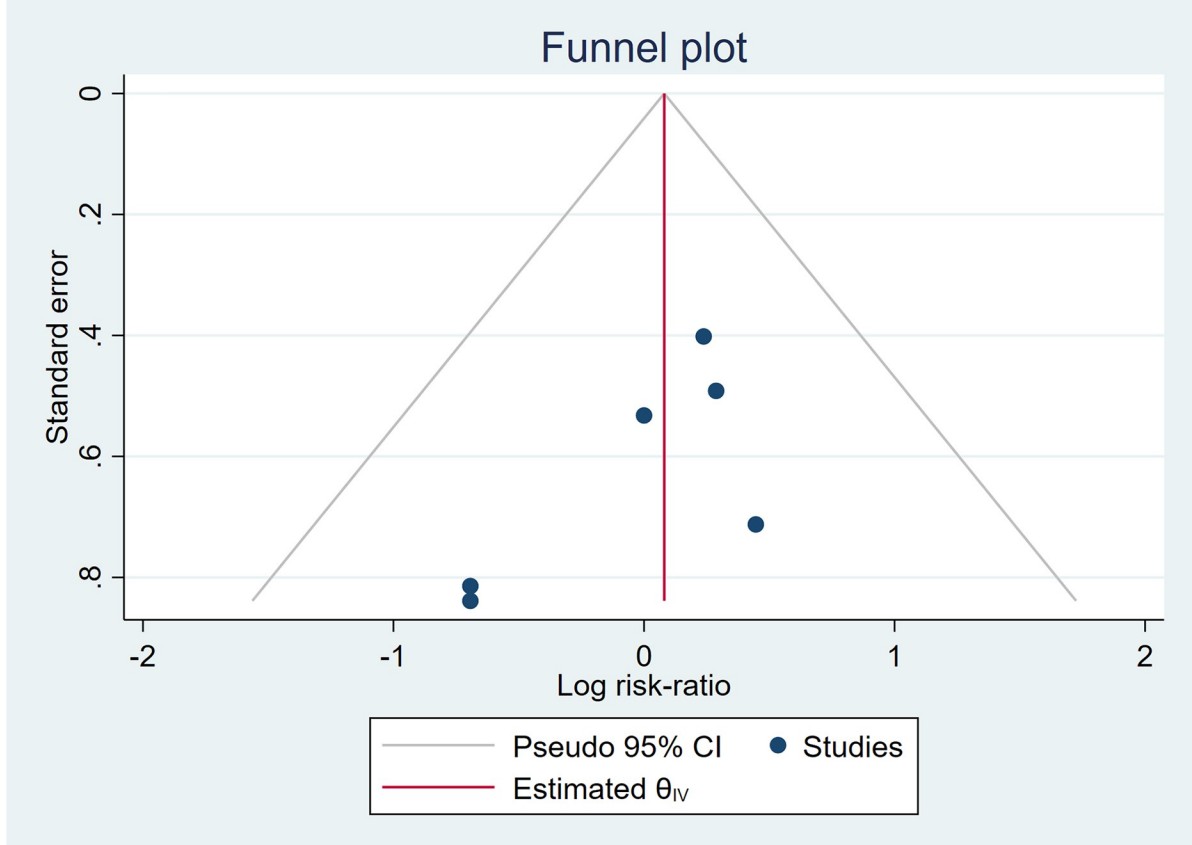

**Fig 3. Meta-analysis of adverse reactions of creatine phosphate sodium in the treatment of viral myocarditis.** Forest plot(A), and funnel plot(B).

notable rise in adverse event rates. These findings suggest CPS's potential utility in VMC treatment, though careful interpretation is advised due to certain study limitations.

Of the nine studies, two examined CPS as a monotherapy, while seven paired it with other treatments such as IVIG, antiviral drugs like ribavirin, cardiac function enhancers like captopril, or metabolic support therapies like sodium fructose diphosphate. CPS was effective in both monotherapy and combination scenarios. However, since this study focused on CPS's overall efficacy in viral myocarditis treatment, it did not compare the standalone versus combination approaches, leaving the potential superiority of combination therapy undetermined. Future studies should prioritize subgroup analyses to clarify these dynamics. In addition, comparative research between IVIG, interferon, and CPS is scarce but essential for evaluating their

relative effectiveness and optimal applications. IVIG primarily targets immune-mediated inflammation, interferon emphasizes antiviral mechanisms, and CPS directly safeguards myocardial cells. Future investigations should focus on head-to-head trials among these treatments to define their respective benefits in different clinical settings.

Variations in the dosage of CPS were observed across the included studies. For comparative analysis, lower doses of CPS were selected as the control group to assess the effects of higher doses. This approach was necessary because no standardized dosage recommendations currently exist for CPS in treating viral myocarditis. Consequently, larger-scale studies are essential to determine the optimal CPS dosage and provide more evidence-based clinical guidance.

The safety profile of CPS was generally favorable among the included studies, with most adverse effects being mild and nonspecific, such as gastrointestinal discomfort and headaches. CPS was largely well-tolerated, with no significant increase in the risk of severe adverse events. This supports its use as a safe adjunctive therapy for VMC. However, due to the limited sample sizes of the included studies, larger-scale, multicenter clinical trials are necessary to establish robust evidence of its long-term safety.

We recognize several limitations in this study. First, the relatively small sample sizes and varying quality among the included studies might have reduced the statistical power and general applicability of the results. Second, potential biases—especially selection bias and performance bias—may have influenced the outcomes, possibly leading to an overestimation of CPS efficacy. Moreover, analyses using funnel plots and Egger's test suggest the presence of publication bias in the cTnI and CK-MB results. This implies that studies with positive findings were more likely to be published, while those with neutral or negative findings might have been overlooked, potentially inflating the perceived effectiveness of CPS. Consequently, the findings of this study should be interpreted with caution. Future research should adopt more rigorous methodologies to minimize bias and ensure comprehensive reporting of both neutral and negative results.

To address the limitations of this study, future research should emphasize the following areas. First, conducting larger-scale, high-quality randomized controlled trials to confirm the efficacy and safety of CPS, thereby improving statistical power and result reliability. Second, comparing standalone CPS therapy with combination treatments to clarify their relative advantages and the appropriate contexts for each approach. In particular, further research is required to explore the potential synergistic effects of combination therapies, such as CPS paired with IVIG or antiviral agents, compared to its standalone use. These analyses should focus on identifying patient subpopulations that derive the most benefit from either approach, optimizing treatment protocols for different clinical scenarios.

## 5.Conclusion

In conclusion, compared to CT therapy alone, CPS therapy may demonstrate efficacy in the treatment of VCM without an increased incidence of adverse reactions and exhibits a favorable safety profile. These findings can provide valuable guidance for the management of VCM. However, due to the limitation of the quantity and quality of the included studies, multi-center, large-sample, rigorously designed and high-quality RCTS are still needed to provide more reliable clinical evidence for this conclusion.

## Supporting information

**S1 Table. PRISMA checklist.**
(DOCX)

**S2 Table. The excluded and included studies were listed in detail.**
(XLS)

**S3 Table. Literature information extraction table & Risk of bias assessment table.**
(XLS)

**S1 Fig. Flow chart of literature screening.**
(TIF)

**S2 Fig. Risk of bias graph.** (a). The judgment of each bias risk item is expressed in percentage in all included studies.(b). Risk of bias summary.
(TIF)

## Acknowledgments

We extend our thanks to all authors for their collaborative efforts. Special thanks go to my wife and all my family members for their silent support in my research work. All data presented in this article are publicly available, and sincere appreciation is extended to all experts who generously contributed to the research data presented in this article. The author takes full responsibility for the content.

## Author Contributions

**Conceptualization:** Su-Fang Chen, Qing-Mei Zhang.

**Data curation:** Li Wang, Xi-Wei Huang, Qing-Mei Zhang.

**Investigation:** Zhi-Yang Wu.

**Methodology:** Su-Fang Chen, Zhi-Yang Wu.

**Writing – original draft:** Li Wang.

**Writing – review & editing:** Li Wang, Xi-Wei Huang, Zhi-Yang Wu, Qing-Mei Zhang.

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
