## [Decision Letter · Decision Letter 0]

5 Nov 2024

PONE-D-24-40066Efficacy and safety of creatine phosphate sodium in the treatment of viral myocarditis：A systematic review and meta-analysisPLOS ONE

Dear Dr. Zhang,

Thank you for submitting your manuscript to PLOS ONE. After careful consideration, we feel that it has merit but does not fully meet PLOS ONE’s publication criteria as it currently stands. Therefore, we invite you to submit a revised version of the manuscript that addresses the points raised during the review process.

Academic editor comments to authors:This systematic review and meta-analysis idea is very interesting, however there are many issues that must be clarified: - The authors ignored the historical role of IVIG in the management of viral myocarditis.... - No comparative studies were mentioned regarding the role of IVIG and other new modalities like interferon or creatine phosphate sodium... - The discussion section is deficient and need to be re-written again.. there is many typo errors that need to be revised precisely

We look forward to receiving your revised manuscript.

Kind regards,

Hany Mahmoud Abo-Haded, MD

Academic Editor

PLOS ONE

2. Please amend either the abstract on the online submission form (via Edit Submission) or the abstract in the manuscript so that they are identical.

5. As required by our policy on Data Availability, please ensure your manuscript or supplementary information includes the following:

Additional Editor Comments:

This systematic review and meta-analysis idea is very interesting, however there are many issues that must be clarified:

- The authors ignored the historical role of IVIG in the management of viral myocarditis....

- No comparative studies were mentioned regarding the role of IVIG and other new modalities like interferon or creatine phosphate sodium...

- The discussion section is deficient and need to be re-written again..

there is many typo errors that need to be revised precisely

Reviewers' comments:

Reviewer's Responses to Questions

**Comments to the Author**

1. Is the manuscript technically sound, and do the data support the conclusions?

Reviewer #1: Yes

Reviewer #2: Yes

Reviewer #3: Yes

2. Has the statistical analysis been performed appropriately and rigorously? 

Reviewer #1: Yes

Reviewer #2: Yes

Reviewer #3: Yes

3. Have the authors made all data underlying the findings in their manuscript fully available?

Reviewer #1: No

Reviewer #2: Yes

Reviewer #3: Yes

4. Is the manuscript presented in an intelligible fashion and written in standard English?

Reviewer #1: Yes

Reviewer #2: Yes

Reviewer #3: Yes

5. Review Comments to the Author

Reviewer #1: Authors has explored the efficacy and safety of CPS use in viral myocarditis by performing metaanalysis. It is interesting research. However, several points require additional clarifications

- In introduction, It was mentioned in the current clinical treatment interferon at first although it is not a standard in management and immunoglobulin are more frequently used although authors did not mention it at all

- In result section under clinical efficacy paragraph, A total of 9 studies is included; however, you cited 10 references (21-24 &26-31) please adjust and remove the extra reference

- I could not find Table 1 among the files available for reviewing, so I could not review the demographics part of the study

- Why data of Meng Wang reference appeared twice in most Forest and Funnel plots figures; please explain.

- The Figure and table legend should be within the main manuscript not in the supporting information section

- Discussion section is limited it would be valuable to discuss how many of the investigated studies used CPS alone and which used CPS in combinations and what was the combinations used.

- Moreover, elaborate on the most recommended dose for CPS as I noticed in inclusion criteria for controls, as authors considered use of low dose CPS to be a control group.

- In discussion, consider writing on the side effects CPS documented in those studies

- Reference 25 has in continuity of it a paragraph of what I think authors roles or conflict of interest part of other paper, please remove it

- Some of references PMID numbers require revision for example references 26,27,28

- It would be valuable to add files detailing the excluded and included studies and additional study data in the Supplementary material section

Reviewer #2: Well written manuscript with proper statistics. The manuscript is technically sound and the well presented . The discussion and the conclusion summarize the important findings. The figures and the statical analysis is appropriate

Reviewer #3: Peer Review of "Creatine Phosphate Sodium in the Treatment of Viral Myocarditis: A Systematic Review and Meta-Analysis"

General Comments

This systematic review and meta-analysis by Wang et al. addresses an important clinical question regarding the efficacy and safety of creatine phosphate sodium (CPS) in the treatment of viral myocarditis (VMC). The authors provide a comprehensive overview of the existing literature and present their findings in a clear, structured manner. However, there are several areas that could benefit from further clarification and refinement to enhance the quality and impact of the manuscript.

Strengths

Relevance: The topic is highly relevant given the increasing recognition of viral myocarditis as a significant health issue. The exploration of alternative treatment options such as CPS is timely and important.

Methodology: The systematic review follows the PRISMA guidelines, and the inclusion of both efficacy and safety outcomes is commendable. The use of established databases for literature search enhances the credibility of the study.

Comprehensive Analysis: The meta-analysis includes a substantial number of patients (1,116), providing a robust data set for analysis. The authors utilize appropriate statistical methods to assess treatment efficacy and safety.

Areas for Improvement

1.Clarity of Results:

While the results are generally well-presented, some sections would benefit from clearer explanations. For example, in the results concerning cardiac troponin I and CK-MB levels, a more detailed description of the clinical significance of the findings would help contextualize the data for readers unfamiliar with these biomarkers.

2.Discussion of Limitations:

The authors acknowledge limitations regarding the sample size and quality of included studies. However, a more in-depth discussion on how these limitations might impact the validity of the conclusions drawn is needed. Specific attention should be given to the potential biases identified in the included studies and how they may affect the reliability of the results.

3.Publication Bias:

The analysis indicates potential publication bias, especially concerning cTnI and CK-MB results. The authors should expand on the implications of this bias and how it may influence the overall conclusions regarding CPS efficacy.

4.Recommendations for Future Research:

While the authors suggest the need for more rigorous studies, they could further detail what specific aspects should be addressed in future research (e.g., study design, patient demographics, dosing strategies).

5.Formatting and Clarity:

Some sections, particularly the methods, could benefit from clearer subheadings to enhance readability. Additionally, the inclusion of flowcharts or diagrams to illustrate the literature selection process could aid comprehension.

Conclusion

Overall, this systematic review and meta-analysis provides valuable insights into the use of CPS for viral myocarditis. With some revisions to enhance clarity, depth of analysis, and discussion of limitations, this manuscript has the potential to make a significant contribution to the field. I recommend acceptance with minor revisions.

Recommendations:

1.Clarify and elaborate on results, particularly in terms of clinical significance.

2.Enhance the discussion of limitations and potential biases.

3.Provide a more detailed outline of future research directions.

4.Improve formatting for better readability.

Thank you for the opportunity to review this manuscript. I look forward to seeing the revised version.

6. PLOS authors have the option to publish the peer review history of their article (what does this mean?). If published, this will include your full peer review and any attached files.

Reviewer #1: No

Reviewer #2: **Yes: **Hala Mounir Agha

Reviewer #3: No

---

## [Author Response · Author response to Decision Letter 0]

15 Dec 2024

List of Responses

Dear Dr Hany Mahmoud Abo-Haded and Reviewers:

Thank you for your letter and for the reviewers’ comments concerning our manuscript entitled “Efficacy and safety of creatine phosphate sodium in the treatment of viral myocarditis：A systematic review and meta-analysis” (ID: PONE-D-24-40066). Those comments are all valuable and very helpful for revising and improving our paper, as well as the important guiding significance to our researches. We have studied comments carefully and have made correction which we hope meet with approval. Revised portion are marked in red in the paper. The main corrections in the paper and the responds to the reviewer’s comments are as flowing:

Responds to Dr Hany Mahmoud Abo-Haded’s comments: 1. Response to comment: The authors ignored the historical role of IVIG in the management of viral myocarditis.

Response: We acknowledge the importance of IVIG in the management of viral myocarditis. In the revised manuscript, we have added in the introduction the historical and current role of IVIG in the treatment of viral myocarditis.

2. Response to comment: No comparative studies were mentioned regarding the role of IVIG and other new modalities like interferon or creatine phosphate sodium.

Response: We have revised the Discussion section to include a comparison of IVIG, interferon, and creatine phosphate sodium, highlighting their mechanisms of action, clinical applications, and limitations.

3. Response to comment: The discussion section is deficient and needs to be re-written again.

Response: The discussion section has been comprehensively rewritten. Based on your feedback as well as that of other reviewers, several modifications have been made, including: a brief explanation of the clinical significance of myocardial biomarkers, a comparative analysis of different therapeutic agents, a discussion of the study's limitations, an analysis of publication bias, and suggestions for future research directions.

4. Response to comment: there is many typo errors that need to be revised precisely.

Response: We feel sorry for our carelessness. In our resubmitted manuscript, the typo is revised. Thanks for your correction.

We appreciate for Editors’ warm work earnestly, and hope that the correction will meet with approval.

Special thanks to you for your good comments.

Responds to the reviewer’s comments:

Reviewer #1:

1. Response to comment: In introduction, It was mentioned in the current clinical treatment interferon at first although it is not a standard in management and immunoglobulin are more frequently used although authors did not mention it at all

Response: We have revised the Introduction to accurately reflect the clinical relevance of IVIG and its historical role in managing viral myocarditis, while clarifying the current understanding and limitations of interferon therapy.

2. Response to comment: In result section under clinical efficacy paragraph, A total of 9 studies is included; however, you cited 10 references (21-24 &26-31) please adjust and remove the extra reference.

Response: We are very sorry for the oversight of the error in the number of articles inserted. We have removed the erroneous references.

3. Response to comment: The problem in Table 1 could not be found.

Response: We have inserted Table 1 into the manuscript.

4. Response to comment: Why data of Meng Wang reference appeared twice in most Forest and Funnel plots figures; please explain.

Response: The reason why the data from Meng Wang’s study appeared twice in most forest and funnel plots is that this study involved different dosage groups of creatine phosphate sodium (CPS) in the treatment of viral myocarditis (VMC). Specifically, the study included two separate experimental groups that received different doses of CPS, each compared to a control group. As each experimental group represents a distinct dose and therapeutic effect, we treated these as independent data points in our analysis, resulting in their separate display in the forest and funnel plots. 

5. Response to comment: The Figure and table legend should be within the main manuscript not in the supporting information section

Response: We have moved all the figures and table legends supporting the information section to the manuscript.

6. Response to comment: Discussion section is limited it would be valuable to discuss how many of the investigated studies used CPS alone and which used CPS in combinations and what was the combinations used.

Response: In the revised manuscript, we have expanded the discussion section. We specified how many studies utilized CPS as a standalone treatment and how many studies employed CPS in combination with other therapies. The overall efficacy of both monotherapy and combination therapy was clarified, while the limitations were highlighted, and directions for future research were outlined.

7. Response to comment: Moreover, elaborate on the most recommended dose for CPS as I noticed in inclusion criteria for controls, as authors considered use of low dose CPS to be a control group.

Response: In our analysis, there were certain variations in the CPS dosages used in the included studies, as there is currently no standardized recommended dosage for CPS in the treatment of viral myocarditis. In dosage comparison studies, lower doses of CPS were sometimes designated as the control group to evaluate the efficacy of higher doses. In these cases, the low-dose group typically received approximately 1 g/day of CPS, while the high-dose group administered increased dosages to assess dose-dependent efficacy. Additionally, in the revised manuscript, we have highlighted in the discussion section that no standard dosage recommendation for CPS in viral myocarditis currently exists. Our findings underscore the need for further research to establish optimal dosage guidelines.

8. Response to comment: In discussion, consider writing on the side effects CPS documented in those studies.

Response: In the discussion section of the revised manuscript, we have added a detailed assessment of adverse effects of CPS in the included studies.

9. Response to comment: Reference 25 has in continuity of it a paragraph of what I think authors roles or conflict of interest part of other paper, please remove it

Response: We have made changes to reference 25, as requested.

10. Response to comment: Some of references PMID numbers require revision for example references 26,27,28

Response: Due to modifications to the content of the original manuscript, the reference numbers have been changed. References 26 and 27 in the original manuscript now correspond to references 30 and 26 in the revised version, respectively.

References 26 and 27 in the original manuscript do not have PMID numbers as they are not indexed in PubMed; therefore, the PMID numbers for these references have been removed. Additionally, reference 28 in the original manuscript was incorrectly cited and has been deleted.

11. Response to comment: It would be valuable to add files detailing the excluded and included studies and additional study data in the Supplementary material section.

Response: we have added detailed files to the Supporting information section, including:1. A table summarizing excluded studies along with the reasons for exclusion (S2 Table).2. A table of all data extracted from the primary research sources for the meta-analysis (S3 Table).3. A table showing the completed risk of bias and quality/certainty assessments for each study or outcome (S3 Table).

We appreciate for Reviewers’ warm work earnestly, and hope that the correction will meet with approval.

Once again, thank you very much for your comments and suggestions.

Reviewer #2:

Response: Thank you for your positive feedback on our manuscript. We greatly appreciate your recognition of the quality of the statistical analysis, the clarity of the presentation, and the relevance of the discussion and conclusion. Your encouraging comments motivate us to continue striving for high standards in our research and writing.

If there are any additional suggestions or areas you think could be further improved, we would be happy to address them.

Reviewer #3:

1. Response to comment: Clarify and elaborate on results, particularly in terms of clinical significance.

Response: we have revised the Results section to provide a more detailed explanation of the clinical significance of cTn I and CK-MB levels.

2. Response to comment: Enhance the discussion of limitations and potential biases.

Response: We have expanded the discussion section to provide a more in-depth analysis of the limitations related to sample size and quality, as well as a detailed examination of how the sample size and quality of the included studies might affect the validity of our conclusions. Additionally, we appreciate your observation regarding the potential publication bias in our analysis, particularly concerning cTnI and CK-MB results. In response, we have further elaborated in the discussion section to clarify the implications of this bias and its potential impact on the conclusions regarding CPS efficacy.

3. Response to comment: Provide a more detailed outline of future research directions.

Response: We have expanded the Discussion section to provide more detailed recommendations for future studies. Specifically, we have highlighted the need for: Rigorous study designs, including larger sample sizes and multicenter randomized controlled trials, to enhance the generalizability and reliability of findings; Stratified analyses based on patient demographics such as age, gender, and disease severity to identify subgroups that may benefit most from CPS treatment; Research focusing on determining the optimal dosing strategies for CPS, including dose-response relationships and comparisons between single and combination therapies.

4. Response to comment: Improve formatting for better readability.

Response: To enhance the clarity and readability of the manuscript, we have made the following modifications: corrected typographical errors and optimized language expression; added clearer subheadings in the Methods section; and uploaded a flowchart illustrating the literature selection process (S1 Fig) in the Supplementary Materials to improve comprehension.

We appreciate for Reviewers’ warm work earnestly, and hope that the correction will meet with approval.

Special thanks to you for your good comments.

---

## [Decision Letter · Decision Letter 1]

31 Dec 2024

Efficacy and safety of creatine phosphate sodium in the treatment of viral myocarditis：A systematic review and meta-analysis

PONE-D-24-40066R1

Dear Dr. Zhang,

We’re pleased to inform you that your manuscript has been judged scientifically suitable for publication and will be formally accepted for publication once it meets all outstanding technical requirements.

Kind regards,

**Prof. Hany M. Abo-Haded, M.D.**

*Academic Editor PLOS ONE*

*Mansoura University, Egypt*

Reviewers' comments:

Reviewer's Responses to Questions

**Comments to the Author**

1. If the authors have adequately addressed your comments raised in a previous round of review and you feel that this manuscript is now acceptable for publication, you may indicate that here to bypass the “Comments to the Author” section, enter your conflict of interest statement in the “Confidential to Editor” section, and submit your "Accept" recommendation.

Reviewer #1: All comments have been addressed

Reviewer #3: All comments have been addressed

2. Is the manuscript technically sound, and do the data support the conclusions?

Reviewer #1: Yes

Reviewer #3: Yes

3. Has the statistical analysis been performed appropriately and rigorously? 

Reviewer #1: Yes

Reviewer #3: Yes

4. Have the authors made all data underlying the findings in their manuscript fully available?

Reviewer #1: Yes

Reviewer #3: Yes

5. Is the manuscript presented in an intelligible fashion and written in standard English?

Reviewer #1: Yes

Reviewer #3: Yes

6. Review Comments to the Author

Reviewer #1: I thank the authors for efficiently and fully addressing all the previously noted comments point by point.

Reviewer #3: (No Response)

7. PLOS authors have the option to publish the peer review history of their article (what does this mean?). If published, this will include your full peer review and any attached files.

Reviewer #1: No

Reviewer #3: No

---

## [Editor Report · Acceptance letter]

14 Jan 2025

PONE-D-24-40066R1 

PLOS ONE

Dear Dr. Zhang, 

I'm pleased to inform you that your manuscript has been deemed suitable for publication in PLOS ONE. Congratulations! Your manuscript is now being handed over to our production team.

Kind regards, 

on behalf of

Professor Hany Mahmoud Abo-Haded 

Academic Editor

PLOS ONE
